# The NICU flora: An effective technique to sample surfaces

**Naomi Sultan[1], Irina Shchors[2], Marc V. Assous[3,4], Maskit Bar-Meir**[4,5] *

**1** Pediatric Department, Shaare-Zedek Medical Center, Jerusalem, IL, United States of America, **2** Neonatal Intensive Care Unit, Shaare-Zedek Medical Center, Jerusalem, IL, United States of America, **3** Microbiology Laboratory, Shaare-Zedek Medical Center, Jerusalem, IL, United States of America, **4** Faculty of Medicine, The Hebrew University, Jerusalem, IL, United States of America, **5** Infectious Diseases, Shaare-Zedek Medical Center, Jerusalem, IL, United States of America

* mbarmeir@gmail.com

**Data Availability Statement:** All relevant data are within the manuscript and its Supporting Information files.

**Funding:** The authors received no specific funding for this work.

## Abstract

### Objective

Environmental surface sampling in healthcare settings is not routinely recommended. There are several methods for environmental surface sampling, however the yield of these methods is not well defined. The aim of the present study is to compare two methods of environmental surface sampling, to characterize the neonatal intensive care unit (NICU) flora, compare it with rates of infection and colonization and correlate it with the workload.

### Design and setting

First, the yield of the swab and the gauze-pad methods were compared. Then, longitudinal surveillance of environmental surface sampling was performed over 6 months,once weekly, from pre-specified locations in the NICU. Samples were streaked onto selective media and bacterial colonies were identified using matrix-assisted laser desorption-ionization time-of-flight (MALDI-TOF).

### Results

The number of colonies isolated using the gauze pad method was significantly higher compared with the swab method. Overall, 87 bacterial species of 30 different bacterial genera were identified on the NICU environmental surfaces. Of these, 18% species were potential pathogens, and the other represent skin and environmental flora. In 20% of clinical cultures and in 60% of colonization cultures, the pathogen was isolated from the infant's environment as well. The number of bacteria in environmental cultures was negatively correlated with nurse/patient ratio in the day prior to the culture.

### Conclusion

The gauze pad method for environmental sampling is robust and readily available. The NICU flora is very diverse and is closely related with the infants' flora, therefore it may serve as a reservoir for potential pathogens.

**Competing interests:** The authors have declared that no competing interests exist.

## Introduction

Healthcare -associated infections (HAI) are responsible for significant morbidity and mortality among neonatal intensive care unit (NICU) patients, and are associated with increased length of stay and healthcare costs [1,2]. It was estimated that environmental contamination can be responsible to up to 20% of HAI [3]. Key pathogens associated with HAIs, such as methicillin- resistant *Staphylococcus aureus* (MRSA), vancomycin-resistant *Enterococcus* (VRE), *Acinetobacter baumannii* and *Clostridium difficile* were shown to persist on surfaces for long periods of time, colonize the hands of healthcare personnel and cause outbreaks [4]. Moreover, in a study of MRSA carriers, it was shown that contact with commonly touched environmental surfaces in patient rooms was equally likely to contaminate the hands of health care providers as was direct contact with the patient [5]. Admission to a room previously occupied by a MRSA or VRE-positive patient increased the odds of acquisition of MRSA and VRE by 40% [6].

Although the contributing role of environmental contamination to HAI is well acknowledged, it is hard to quantify the risk of transmission. This risk depends on qualities of the pathogen (e.g. ability to survive on surfaces), cleaning practices and exceeding a certain threshold of contamination [6]. Environmental surface sampling in healthcare settings is not routinely recommended, and is indicated for research, epidemiologic investigation or for purposes of quality assurance of cleaning practices [7]. The selection of appropriate sample technique is necessary in order to obtain meaningful results. Effective sampling of surfaces always requires moisture, and the guidelines of the Center for Disease Control (CDC) detail several methods for environmental-surface sampling [7]. However, during a past outbreak in our NICU caused by carbapenem-resistant *Acinetobacter baumannii* (CRAB), we failed to demonstrate the role of the environment as a reservoir. At the time we used moistened swabs, rubbed them against frequently touched surfaces and processed them per guidelines, but were not able to isolate CRAB. New patients continued to acquire CRAB colonization, despite our intensive efforts to enforce strict hand hygiene and proper terminal cleaning, and despite strict patient isolation, cohorting and separation of teams taking care of colonized and uncolonized patients. Only complete physical separation into two units ('clean' and 'colonized/'exposed') ended the outbreak (unpublished data). This experience led us to question the yield of our sampling method. The aim of the present study is to compare two methods of environmental surface sampling, to characterize the NICU flora,compare it with rates of infection and colonization and correlate it with the workload (e.g. number of patients and nurse/patient ratio).

## Methods

### Sample collection and processing

Environmental samples were obtained from surfaces in patient rooms while occupied. First, we compared two sample methods as outlined in the CDC guideline[7]: *a. swab method-* each site was sampled using a sterile cotton swab pre-moistened with 0.9% Nacl for injection. The swab was rolled back and forth over each surface three times to ensure that all sides of the swab made contact with the surface and that maximal surface area was covered. *b. gauze pad method-* a sterile 7.5X7.5 cm gauze pad was moistened with Mueller-Hinton broth and rubbed against the surface in the same manner as the swab. Each surface was sampled first using the swab and then the gauze method, 3 samples were also obtained in the opposite order (gauze first and then swab).

Both the swabs and the gauze pads were incubated overnight in 10 mL Mueller-Hinton broth at 37˚C. Samples with visible turbidity were streaked onto blood and MacConkey plates, as well as on selective media for resistant organisms: MRSA, CRAB and carbapenem-resistant

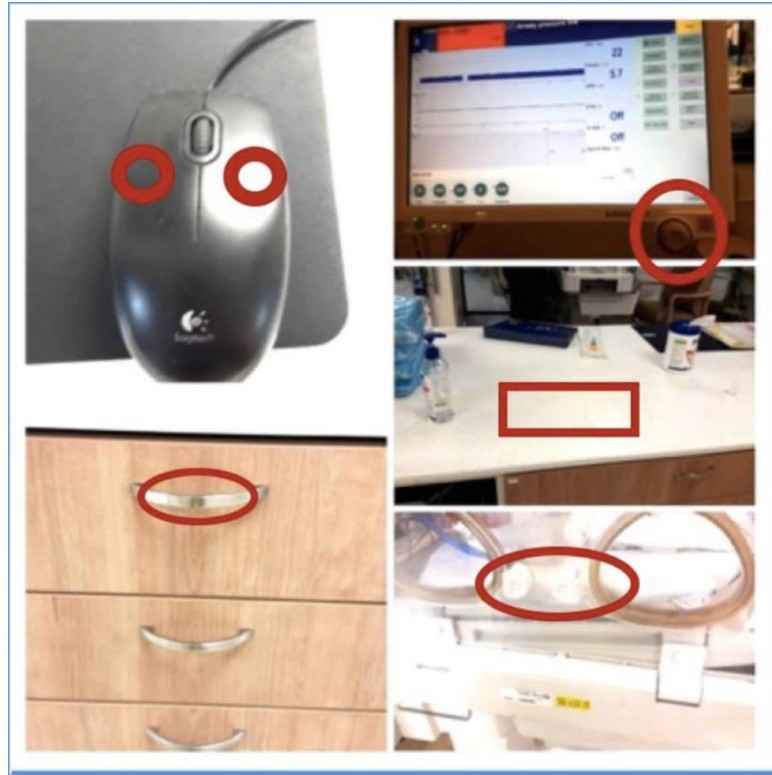

**Fig 1. Frequently touched surfaces in the neonatal intensive care unit from which surface cultures were obtained: Computer mouse, the monitor screen knob, drawer handles, incubator handles and the surface of the counters.**

enterobacteriaceae (CRE). Colonies were identified by morphology. A representative colony was identified using matrix-assisted laser desorption ionization time-of-flight (MALDI-TOF) performed with a MicroFlex LT system (Bruker Daltonics) tabletop mass spectrometer using the manufacturer's suggested settings. Captured spectra were analyzed using MALDI Biotyper automation control and Bruker Biotyper 2.0 software (Bruker Daltonics, Bremen, Germany).

A negative control (swab or gauze) was included in each round. Frequently touched surfaces for sampling were chosen after an observation performed by two of the authors (I.S, M. BM), and included the computer mouse, the monitor screen knob, drawer handles, incubator handles and the surface of the counters (Fig 1).

The better-performing method (swab vs. gauze) was chosen for the subsequent stage of the study. Longitudinal surveillance of environmental surface samples was performed over 6 months. Once weekly, 10 samples were obtained -half were obtained from surfaces in a "clean" room (e.g room of non-colonized infants) and half from a room of infants in contact isolation (infants colonized with extended-spectrum beta-lactamase- ESBL- producing enterobacteriaceae or MRSA), for a total of 240 samples. The same procedures described in the first stage were used here as well.

**Routine sampling, isolation policy and cleaning procedures.**   Routine surveillance cultures to detect colonization among the infants were performed in our NICU once a week; nasopharyngeal swabs for MRSA screening and rectal swabs for ESBL and CRE. CRAB could be identified in both rectal and nasopharyngeal cultures. Infants with positive surveillance cultures were placed in contact isolation. Cohorting of infants based on their colonization status was attempted. For MRSA colonized infants, decolonization was performed using intranasal

mupirocin ointment 2% twice daily and 2% chlorhexidine washes for 7 days [8]. Routine cleaning of counter' surfaces and drawer handles in the NICU was performed twice a day using 1000 ppm sodium hypochlorite. Other areas sampled in this study (computer mouse, monitor screen knobs, incubator handles) were cleaned daily with quaternary ammonium wipes.

### Data analyses

All bacterial isolates from clinical and surveillance cultures in NICU patients during the study period were recorded. Isolates growing from more than one site were counted once. Correlation was examined between pathogens grown on clinical cultures in the two weeks prior to obtaining the environmental cultures. The number of patients and the average number of nurses in each shift was recorded daily as well.

The gauze pad method was found to support a large number of bacteria, therefore we did not perform quantitative analysis of the cultures, but rather analyzed them qualitatively. The number of different bacterial strains was recorded and correlated with the results of clinical and surveillance cultures during the prior week.

Correlation between the number of bacterial strains in the environmental cultures and the nurse/patient ratio in the day prior to the culture was done using a linear regression with the number of bacterial strains as the dependent variable.

**Ethics.** This study was approved by the SZMC institutional Helsinky committee. Written informed consent was waived, since all clinical data were de-identified.

## Results

### Swab vs. gauze-pad method

Overall, 30 surface cultures were obtained using cotton swabs pre-moistened with 0.9% Nacl and 30 using gauze pad moistened with Mueller-Hinton broth (5 samples in each method on three separate occasions). The number of colonies isolated using the gauze pad method was significantly higher (p = 0.0001; Fig 2). A triplicate of samples obtained with gauze first and

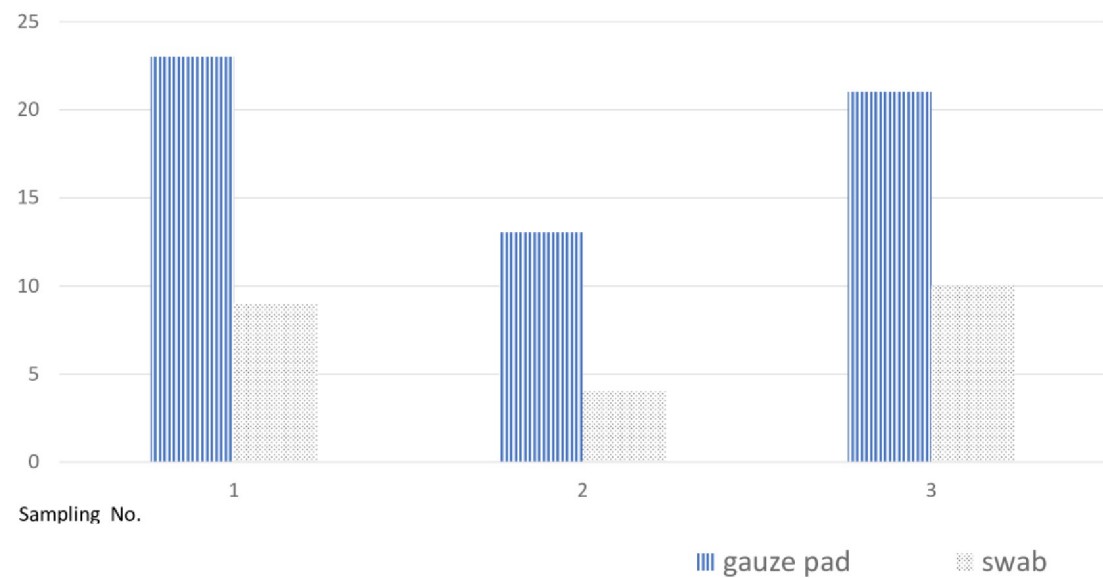

**Fig 2. Number of colonies isolated in environmental cultures obtained by swab moistened with 0.9% Nacl (dotted bars) vs. gauze pad moistened with Mueller-Hinton broth (vertical stripes).**

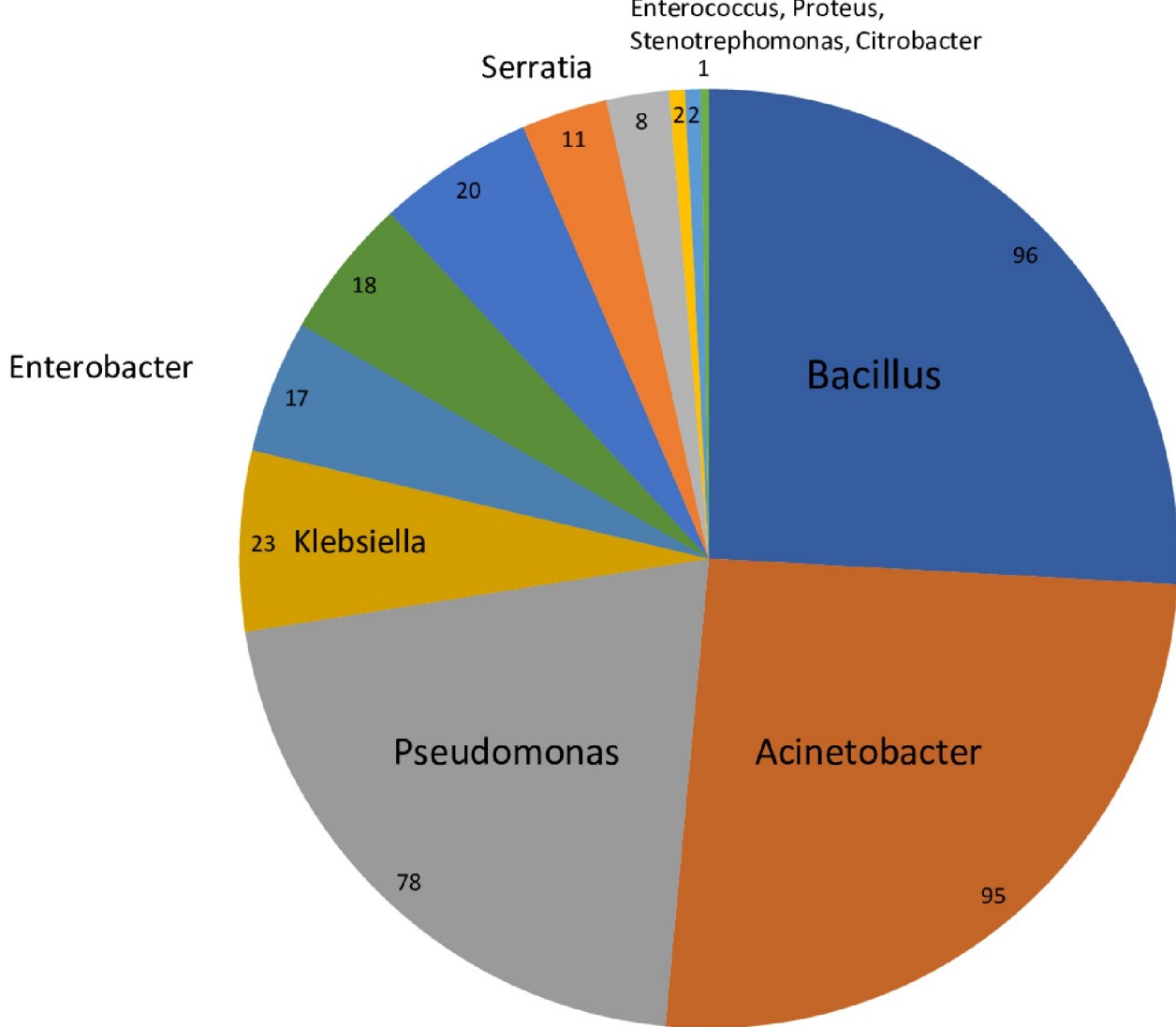

**Fig 3. Bacterial genera isolated from surface cultures in the neonatal intensive care unit.**

then the swab yielded similar results. The gauze pad method was therefore selected for further analysis.

**Analysis of the NICU flora.** We identified 87 different bacterial species belonging to 30 genera growing on the environmental surface cultures (Fig 3, S1 File). *Bacillus* species, including *Bacillus cereus* was the most frequently isolated genus. Other frequently isolated species included skin flora, such as *Acinetobacter lwoffi* or *Microccus luteus*, and environmental bacteria such as *Pseudomonas stutzeri*. Interestingly, we have also identified *Massilia timonae*, an infrequent human pathogen.

Potential (frequent) pathogens were identified in 16 cases (18%). The most frequent pathogens isolated were gram -negative bacteria. Enterobacteriaceae were the most common: *Klebsiella pneumoniae* followed by *Enterobacter cloacae* and *E.coli*. Other pathogens recovered from the NICU environment included *Serratia marcescens*, *Enterococcus fecalis* and *Staphylococcus aureus*. Cultures obtained from the incubator handles yielded the largest number of bacteria,

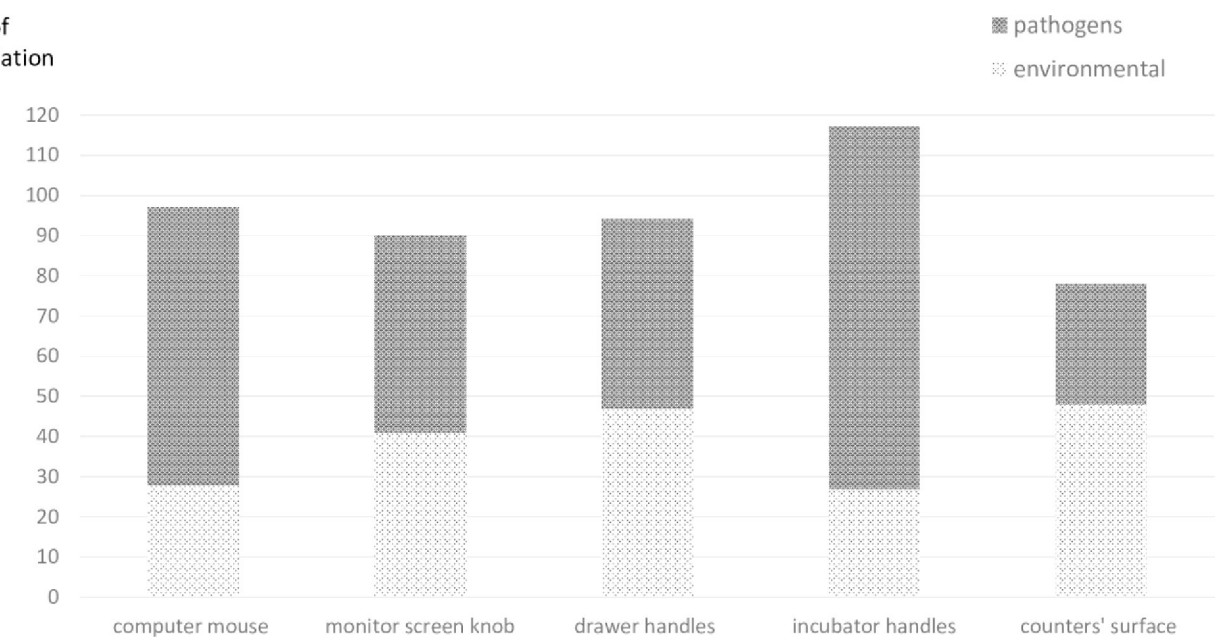

**Fig 4. The number of colonies grown from environmental surface cultures by site of culture and environmental vs. pathogenic bacteria.**

followed by the computer mouse, the drawer handles, the monitor screen knob and the counters' surface. Fig 4 shows the number of colonies grown from each site.

**Correlation between environmental cultures and clinical samples, colonization and workload.** Of the 40 positive clinical cultures recorded over the study period, in 8 (20%) cases the pathogen was isolated in the environment as well. Table 1 shows the correlation in these cases. Of note, sputum cultures were most frequently associated with environmental contamination.

In 3 of 5 infants diagnosed as carriers of resistant pathogens, the same pathogen was isolated in surface cultures from the infants' room (1 MRSA, 2 CRE).

The number of bacteria in environmental cultures was negatively correlated with nurse/patient ratio in the day prior to the culture, although it did not reach statistical significance (B = -27, 95%CI:-60,-4.4, p = 0.08).

**Table 1. Correlation of time and site between clinical and environmental cultures during the study period.**

| Case # | Pathogen | Site of environmental culture | Source of clinical culture | Time between environmental and clinical culture (days) |
|---|---|---|---|---|
| 1 | *Bacillus cereus* | monitor screen knob | blood | 3 |
| 2 | *Acinetobacter pittii* | incubator handles | sputum | 3 |
| 3 | *Proteus mirabilis* | incubator handles | eye discharge | 2 |
| 4 | *Klebsiella oxytoca* | computer mouse | sputum | 1 |
| 5 | *Klebsiella pneumoniae* | monitor screen knob, incubator handles, drawer handles | sputum | 5 |
| 6 | *Klebsiella oxytoca* | monitor screen knob | sputum | 7 |
| 7 | *Klebsiella pneumoniae* | monitor screen knob, incubator handles, drawer handles, computer mouse | sputum | 12 |
| 8 | *Enterobacter cloacae* | computer mouse | sputum | 10 |

## Discussion

Fomites are defined as objects that can serve as vehicles for transmission of an infectious agent [7]. The role of fomites (e.g. patient care items or environmental surfaces) as reservoir or a source of nosocomial infections is difficult to define. Factors such as sampling technique, levels of hand hygiene and environmental cleaning, as well as different survival capabilities of pathogens, make it difficult to assess the role of fomites in nosocomial outbreaks.

Here, we show that the gauze pad method had a much better yield compared with the swab method. The swab is cheap and easy to use, however the gauze covers larger surface and is moistened with broth, therefore we found an abundant growth of many different genera of bacteria. Many of the species growing in the NICU environmental cultures represent non-pathogenic bacteria. To that end, the gauze method may give much information without clinical significance. Culturing the broth-moistened gauze on selective media may take advantage of the high sensitivity of the method as well as save the processing of non-pathogenic bacteria.

The gauze method demonstrated the high diversity of the NICU flora. Methods of high throughput sequencing found an average of 100 bacterial genera on every surface cultured in the NICU [9]. Applying these methods on various NICU surfaces provide a broad understanding of the different taxa of bacteria and fungi that constitute the NICU microbiota and show how cleaning changes this ecosystem [10]. However, these sequencing methods could not provide a resolution at the strain level, therefore could not differentiate pathogens from closely-related non-pathogens.

Using the gauze method we could demonstrate the role of environmental contamination for cross-transmission of pathogens. Almost a fifth of the bacterial strains in the environmental cultures were potential pathogens. In 20% of cases with positive clinical cultures, as well as in 3 of 5 cases with positive surveillance cultures, the pathogen was isolated from the patient's environment. However, our study design does not address the temporality (i.e., whether the environment is contaminated before or after patient infection/colonization).

*B.cereus* was the most frequently isolated potential pathogen in the NICU environmental cultures. We have recently described a cluster of severe healthcare-associated *B.cereus* infection in our NICU, likely associated with construction- related dust [11]. The role of contaminated fomites as vehicles of *B.cereus* nosocomial infections and its seasonality were demonstrated repeatedly [12–15]. Routine surveillance of environmental cultures may enable early recognition of *B.cereus* environmental contamination and support timely infection control interventions.

Frequent bacterial species found in the NICU environment are those that consist the skin flora, such as *Acinetobacter lwoffi*, or those originating from the environment, such as *Pseudomonas stutzeri or Pantoeae*. These pathogens are sometimes encountered in blood cultures and are frequently considered contaminants, but can also be associated with true infections and even outbreaks in the NICU [16,17]. Interestingly, we also isolated *Massilia timonae*, an infrequent human pathogen that was not described so far as causing infections in premature infants [18].

We found that sputum cultures were most frequently associated with environmental contamination. Contaminated respiratory devices, such as humidifiers, nebulizers and suction apparatus, were frequently associated with nosocomial outbreaks. In these cases, pathogen transmission could occur through healthcare personnel, aerosolization into room air or cross-transmission through fomites [19].

The largest number of pathogens was found on the incubator handles. This is probably the most frequently touched surface in the NICU and is in the closest vicinity to the baby's flora, therefore deserves special attention during cleaning routines.

Finally, we found a negative correlation between the workload in the NICU, as evaluated by nurse/patient ratio, and the bacterial load. Overcrowding and understaffing were repeatedly shown to play a central role in nosocomial outbreaks in the NICU [20,21].

Out study has several limitations. First, we could not ensure that the time interval from the twice-daily cleaning of NICU surfaces to obtaining the surface cultures is similar (cultures could theoretically be obtained right before or right after the cleaning, which could influence the culture results). Second, as noted above, the temporality of the patients' vs. environmental pathogens cannot be determined (e.g. whether the environment is contaminated before or after patient infection/colonization). Third, there are no standards that allow meaningful comparison of contamination levels and "permissible" levels of microbial contamination are undefined. Moreover, we could not evaluate levels of hand hygiene, environmental cleaning or disinfection.

In summary, we describe a powerful method of sampling various surfaces in the patient vicinity and show the diversity of the NICU environmental flora and the potential role of fomites in cross-transmission. There are currently no guidelines that endorse routine environmental sampling in the NICU, however incorporating such sampling into the infection control efforts in the context of an outbreak may add a useful monitoring tool to the cleaning routines and aid in breaking the cross-transmission cycles.

## Supporting information

**S1 File. Type of bacteria.**
(XLSX)

## Author Contributions

**Conceptualization:** Irina Shchors, Marc V. Assous, Maskit Bar-Meir.

**Data curation:** Marc V. Assous, Maskit Bar-Meir.

**Investigation:** Naomi Sultan.

**Methodology:** Irina Shchors, Marc V. Assous.

**Resources:** Maskit Bar-Meir.

**Supervision:** Irina Shchors, Marc V. Assous, Maskit Bar-Meir.

**Writing – original draft:** Maskit Bar-Meir.

**Writing – review & editing:** Marc V. Assous.

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
