## [Decision Letter · Decision Letter 0]

9 Aug 2021

PONE-D-21-08619

The NICU flora: An Effective Technique to Sample SurfacesThe NICU flora: An Effective Technique to Sample Surfaces

PLOS ONE

Dear Dr. Bar-Meir

Thank you for submitting your manuscript to PLOS ONE. After careful consideration, we feel that it has merit but does not fully meet PLOS ONE’s publication criteria as it currently stands. Therefore, we invite you to submit a revised version of the manuscript that addresses the points raised during the review process.

We look forward to receiving your revised manuscript.

Kind regards,

Mohammad Mehdi Feizabadi, PhD

Academic Editor

PLOS ONE

Additional Editor Comments (if provided):

Please check the comments given by two expert reviewers. Your manuscript needs minor changes

Journal Requirements:

Reviewers' comments:

Reviewer's Responses to Questions

**Comments to the Author**

1. Is the manuscript technically sound, and do the data support the conclusions?

Reviewer #1: Yes

Reviewer #2: Yes

2. Has the statistical analysis been performed appropriately and rigorously? 

Reviewer #1: Yes

Reviewer #2: Yes

3. Have the authors made all data underlying the findings in their manuscript fully available?

Reviewer #1: Yes

Reviewer #2: Yes

4. Is the manuscript presented in an intelligible fashion and written in standard English?

Reviewer #1: Yes

Reviewer #2: Yes

5. Review Comments to the Author

Reviewer #1: Thank you for your consideration inviting me to review your study. A few question:

1. It showed that swab sampling was done before gauze-pad, will it make any differences if you switched the surface sampling order; the gauze-pad first then swab?

2. From a positive environmental culture result, is there a relationship with the condition of the previous patient who occupied the same incubator?

3. Can you tell me about your routine incubator cleaning? Is there a decrease in environmental culture (since the environmental culture is carried out several times with a certain time span)

4. How is the condition of the previous patient with clinical sepsis and positive culture showing HAI, are there new patients who are admitted with symptoms of sepsis?

5. How to inoculate in a culture medium with a gauze sampling method? Won't it be easier to inoculate if you had it done with swab?

Reviewer #2: I have two questions about this article...

Did the gauze-pad method not have a higher detection power in identifying the type of bacteria than the swab method?

If yes It is better to give more details

Was MRSA strain isolated in this study?

6. PLOS authors have the option to publish the peer review history of their article (what does this mean?). If published, this will include your full peer review and any attached files.

Reviewer #1: No

---

## [Author Response · Author response to Decision Letter 0]

9 Sep 2021

Reviewer #1: 

1. It showed that swab sampling was done before gauze-pad, will it make any differences if you switched the surface sampling order; the gauze-pad first then swab?

The cultures were obtained at the exact same time (e.g probably no time for real change in the microbial flora), so we did not think it should matter, but per the reviewer comment we have repeated a triplicate of gauze pad first then swab from one of the NICU counters. The results were quite similar with an average of 20 colonies in the gauze-pad cultures and 5 in the swab. We have added it to methods and results. 

2. From a positive environmental culture result, is there a relationship with the condition of the previous patient who occupied the same incubator?

It is an interesting question, unfortunately I do not think we know the answer. The infants, especially the extremely premature, may stay in an incubator for weeks. Our routine is to replace the incubator every 7-10 days. Although we have a software that detects “neighbors” – patients who stay in the same room at the same time- we do not have a good way to “tag” the incubators, and connect them with a certain patient. 

3. Can you tell me about your routine incubator cleaning? Is there a decrease in environmental culture (since the environmental culture is carried out several times with a certain time span)

Between patients, the incubator is cleaned with a 0.1% (1000 ppm) chlorine solution. While the baby is inside the incubator, the outer surfaces are cleaned every shift with alcohol wipes. The inner surface is not cleaned, but we replace the incubator every 7-10 days. As for the second question- The gauze pad method was found to support a large number of bacteria , therefore we did not perform quantitative analysis of the cultures , but rather analyzed them qualitatively. Therefore we cannot comment on a quantitative decrease, but qualitatively – the diversity of the bacteria over the weeks of the study did not show a decrease.

4. How is the condition of the previous patient with clinical sepsis and positive culture showing HAI, are there new patients who are admitted with symptoms of sepsis?

I wasn’t sure if the reviewer is asking about the patient with the blood culture positive for B.cereus? This patient has fully recovered. We correlated the clinical bacteria during the study period with the environmental bacteria (Table 1), there was some correlation with clinical cultures as noted in the manuscript, but generally speaking we did not have any outbreak or an unusual rate of infections during this period. 

5. How to inoculate in a culture medium with a gauze sampling method? Won't it be easier to inoculate if you had it done with swab?

After sampling, the gauze (or the swab) are introduced into a collector 50 ml tube. Then, 10 ml of Muller-Hinton broth is added and incubated overnight. The swab follows the same procedure. Then, only the broth ,where the gauze or swab were incubated, is cultured. Therefore, both inoculations are in fact technically the same, but then the yield of the swab is lower, as we show in this work. 

Reviewer #2: 

 Did the gauze-pad method not have a higher detection power in identifying the type of bacteria than the swab method?

The gauze-pad method has a better yield in supporting the growth of bacteria. Once grown, the identification of type of bacteria was done by colony morphology and matrix-assisted laser desorption ionization time-of-flight (MALDI-TOF). Therefore, the gauze helps supporting the growth but does not influence the identification 

Was MRSA strain isolated in this study?

We had 4 S.aureus strains, 1 of these MRSA, in a room of a patient colonized with MRSA (see results section).

---

## [Editor Report · Decision Letter 1]

13 Sep 2021

The NICU flora: An Effective Technique to Sample Surfaces

PONE-D-21-08619R1

Dear Dr.Maskit Bar-Meir

We’re pleased to inform you that your manuscript has been judged scientifically suitable for publication and will be formally accepted for publication once it meets all outstanding technical requirements.

Kind regards,

Mohammad Mehdi Feizabadi, PhD

Academic Editor

PLOS ONE
---

## [Editor Report · Acceptance letter]

15 Sep 2021

PONE-D-21-08619R1 

The NICU flora: An Effective Technique to Sample Surfaces 

Dear Dr. Bar-Meir:

I'm pleased to inform you that your manuscript has been deemed suitable for publication in PLOS ONE. Congratulations! Your manuscript is now with our production department. 

Kind regards, 

on behalf of

Dr. Mohammad Mehdi Feizabadi 

Academic Editor

PLOS ONE